# Cytotoxicity Assessment of a New Design for a Biodegradable Ureteral Mitomycin Drug-Eluting Stent in Urothelial Carcinoma Cell Culture

**DOI:** 10.3390/polym14194081

**Published:** 2022-09-29

**Authors:** Federico Soria, Luna Martínez-Pla, Salvador D. Aznar-Cervantes, Julia E. de la Cruz, Tomás Fernández, Daniel Pérez-Fentes, Luis Llanes, Francisco Miguel Sánchez-Margallo

**Affiliations:** 1Jesus Uson Minimally Invasive Surgery Centre Foundation, Endoscopy-Endourology Department, 10071 Cáceres, Spain; 2Biotechnology, Genomics and Plant Breeding Department, Instituto Murciano de Investigación y Desarrollo Agrario y Ambiental (IMIDA), La Alberca, 30150 Murcia, Spain; 3Urology Department, Morales Meseguer University Hospital, 30008 Murcia, Spain; 4Urology Department, Santiago de Compostela University Hospital, 15706 Santiago de Compostela, Spain; 5Urology Department, Getafe University Hospital, 28905 Madrid, Spain

**Keywords:** ureteral stent, biodegradable stent, cell viability assessment, chemotherapy, UTUC

## Abstract

Urothelial tumour of the upper urinary tract is a rare neoplasm, but unfortunately, it has a high recurrence rate. The reduction of these tumour recurrences could be achieved by the intracavitary instillation of adjuvant chemotherapy after nephron-sparing treatment in selected patients, but current instillation methods are ineffective. Therefore, the aim of this in vitro study is to evaluate the cytotoxic capacity of a new instillation technology through a biodegradable ureteral stent/scaffold coated with a silk fibroin matrix for the controlled release of mitomycin C as an anti-cancer drug. Through a comparative study, we assessed, in urothelial carcinoma cells in a human cancer T24 cell culture for 3 and 6 h, the cytotoxic capacity of mitomycin C by viability assay using the CCK-8 test (Cell counting Kit-8). Cell viability studies in the urothelial carcinoma cell line confirm that mitomycin C embedded in the polymeric matrix does not alter its cytotoxic properties and causes a significant decrease in cell viability at 6 h versus in the control groups. These findings have a clear biomedical application and could be of great use to decrease the recurrence rate in patients with upper tract urothelial carcinomas by increasing the dwell time of anti-cancer drugs.

## 1. Introduction

Urothelial carcinomas are the sixth most common tumour in developed countries [1]. These carcinomas are mainly located at the bladder level (90–95%), and only 5–10% correspond to upper tract urothelial carcinoma (pyelocaliceal cavities and ureter) (UTUC). The highest incidence of UTUC is in the age group between 70 and 90 years, and UTUC is diagnosed in men twice as often as in women [2,3]. In contrast to bladder cancer, two-thirds of patients at the time of their diagnosis of UTUC have invasive musculo-infiltrative disease [4].

The gold standard for UTUC treatment is radical nephroureterectomy (RNU), which is mainly for tumours classified as high-risk. However, one-third of UTUC cases may benefit from nephron-sparing surgery: those that are classified as low-risk according to the current European Association of Urology (EAU) guidelines [5]. In addition, selected patients with high-risk UTUC, such as patients with a solitary kidney or advanced chronic kidney disease, patients who refuse RNU, or patients with bilateral UTUC, are also candidates for endoscopic management, as they are not candidates for nephrectomy [6]. Unfortunately, unlike the well-established protocol for bladder urothelial tumours, the instillation of chemotherapy as an adjuvant treatment for UTUC in the upper urinary tract following endoscopic treatment of UTUC by ureteroscopic laser ablation is uncommon [7]. The aim of chemotherapy instillation is the reduction of the recurrence rate of UTUC, as evidenced by some authors in the scientific literature [8]. However, due to the particularities of the upper urinary tract, intracavitary instillation is very challenging. This is mainly due to the small volume of the upper urinary tract as well as to the inadequate drug dwell time of topical agents [7].

Current research seeks to develop new upper urinary tract chemotherapy delivery systems that can overcome the limitations of traditional techniques, mainly using drug-eluting stents and thermosensitive polymers [7,9]. The most recent advance has been UGN-101, a novel formulation of mitomycin C (MMC) containing a reverse thermal gel designed to increase urinary dwell time and thereby the efficacy of treatment. It showed a 56% complete response rate after 12 months in its first clinical trial, but with significant related side effects, including 44% of patients experiencing ureteral stricture [10]. However, the development of chemotherapy-eluting ureteral stents has not yet made enough progress due to difficulties in drug-delivery control through the stent coating. In this regard, one of the possible solutions that could help to overcome some of the biggest shortcomings in stent technology may be silk fibroin (SF) from *Bombyx mori* as a carrier and delivery system for drug-eluting stents due to its biocompatibility and easy processing [11]. SF’s features make this protein a great candidate for biomaterial-controlled release technology [11].

For these reasons, the aim of our in vitro study is to assess cell viability in a urothelial carcinoma cell culture using a new design for biodegradable ureteral stents coated with SF for the controlled release of mitomycin C (BraidStent-SF-MMC) for topical instillation in the adjuvant treatment of UTUC. We need to confirm whether the MMC released by the stent maintains its cytotoxic capacity after embedding in the SF polymeric matrix.

## 2. Materials and Methods

### 2.1. BraidStent-SF-MMC

#### 2.1.1. Materials for Biodegradable Ureteral Stent Preparation

To perform the in vitro study, 24 fragments of a 10 mm long BraidStent^®^ (JUMISC, Cáceres, Spain) biodegradable ureteral stent were made by combining biodegradable polymers and copolymers—Glycomer™ 631 and polyglycolic acid (PGA), at a ratio of 54% and 46%, respectively [12,13,14]. Two 0.17 mm thick threads of Glycomer™ 631 and two 0.14 mm thick threads of PGA, all 10 mm long, were used to manufacture the stent. The threads were then braided together to constitute the central core of the stent. These 24 stent fragments all underwent coating with SF, and 12 of them subsequently underwent the addition of a chemotherapeutic agent widely used for the intracavitary instillation in the UTUC, namely MMC [8].

#### 2.1.2. Materials for Stent SF and MMC Coating

Cocoons of *Bombyx mori* were obtained from worms reared in the sericulture facilities of the Imida Instituto Murciano de Investigación y Desarrollo Agrario y Ambiental (IMIDA), Biotechnology, Genomics and Plant Breeding Department (La Alberca, Murcia, Spain). Cocoons were chopped up and boiled in 0.02 M Na_2_CO_3_ for 30 min to eliminate the sericin. Then, the raw SF was rinsed with distilled water and dried at room temperature for 3 days. Subsequently, SF was dissolved in 9.3 M LiBr (Acros Organics) for 3 h at 60 °C, yielding a 20% weight per volume (*w*/*v*) dissolution that was dialysed against distilled water for 3 days (Snakeskin Dialysis Tubing 3.5 KDa MWCO, Thermo Scientific, Rockford, IL, USA), with eight total water changes (at 4 °C) [15]. The resultant 7–8% *w*/*v* SF solution was recovered and used for the preparation of the coated stents, adjusting the concentration to 7% *w*/*v* before use [16,17].

The coating of the BraidStent-SF-MMC stent was carried out with powdered 70 mg pure MMC (Mitomycin CRS, Sigma-Aldrich, Darmstadt, Germany). To this end, the concentration employed for both the fibroin and methanol solutions was 10 mg/mL, and 10 dip-coating cycles were performed by dissolving the drug (10 mg/mL MMC) for 30 min under orbital agitation by stirring at 120 rpm. These two solutions were used to alternately coat the stent fragments by dipping them in the first SF solution for 5 s and then in the methanol solution for 5 s, allowing them to dry for 1 min before repeating the procedure. The stents were completely stable in room air. However, the MMC coating needs special care (i.e., it must be packaged in lightproof packaging) [17].

In previous studies by our research group, stent coating characterisation was performed to determine whether there was adequate SF coating on the BraidStent-SF-MMC surface by adding a fluorescent aqueous marker (sulforhodamine B). The results of the fluorescence microscopy study confirmed the homogeneity of the SF coating on the stent surface [17].

#### 2.1.3. Determination and Assessment of the MMC Release from BraidStent-SF-MMC

To determine MMC release, we tested 10 fragments of BraidStent-SF-MMC that were 10 mm in length and coated as described above in artificial urine (AU) (Human Synthetic Urine, BioIVT, West Sussex, UK). The stent fragments were incubated in an orbital shaker incubator under mimicked biological conditions (36.5 °C with 5% CO_2_ at 90 rpm). The concentration of MMC released in the AU at 3 and 6 h was determined by HPLC-DAD. Urine from each follow-up was replaced and analysed. The permeability and release kinetics of the MMC depends on the SF coating and is related to the percentage of beta-sheet structure, which is controlled by dipping the SF solution in methanol [17].

The HPLC-DAD method is an isocratic method with an acetonitrile mobile phase. Ultrapure water that was 80:20 volume per volume (*v*/*v*) at a flow rate of 1 mL/min was used, and separation was performed on a LUNA C18 using 250 mm, 4.6 mm, 5 µm columns at 30 °C. The MMC was detected with a diode array detector (DAD) at 365 nm (1260 Infinity II Prime LC System, Agilent Technologies, Santa Clara, CA, USA) [17].

### 2.2. T24 Cell Culture Line

Urothelial carcinoma cells from human bladder cancer T24 cells were used as a model cell line (EP-CL-0227, BioNova científica^®^, Madrid, Spain) [18,19]. This is a tumour cell line from human transitional bladder cell carcinoma and is frequently used to assess the cellular cytotoxicity of chemotherapeutics in the urinary tract in vitro [18,19,20].

Human T24 cells were seeded in plates containing McCoy’s 5a (Thermo Fisher Scientific^®®®^, Madrid, Spain) and supplemented with 10% foetal bovine serum (FBS), 1% penicillin/streptomycin, and 1% glutamine and were incubated at 37 °C at 95% relative humidity and 5% CO_2_. Cells were maintained at 37 °C in a humidified 5% CO_2_ atmosphere for 48 h.

### 2.3. In vitro Cytotoxicity of the BraidStent-SF-MMC

At 80% confluence, adherent cells were detached with Trypsin EDTA solution (Lonza Bioscience, Pontevedra-Spain) and seeded in four 24-well plates at a final concentration of 20,000 cells per well. Once adherent, four groups of studies were performed. In group 1 (G1), cell viability was assessed after the instillation of pure mitomycin C was added at a concentration of 0.66 mg/mL, the concentration recommended for the adjuvant treatment of UTUC in medical practice [8]. Group 2 (G2), the BraidStent-SF-MMC stent group, is the subject of the current study. The negative controls used were the BraidStent-SF without MMC coating (group 3, G3) and T24 cells in an MMC-free medium (group 4, G4). The sample size was six samples per group. All of these groups were assessed at baseline (T0), at 3 h (T3), and at 6 h (T6) (Table 1; Figure 1).

Mitomycin C cytotoxicity was assessed by a viability assay using CCK-8 (Cell Counting Kit-8, Boster Biological Technology, Pleasanton, CA, USA). The protocol was carried out according to the manufacturer’s recommendations, and the absorbance at 450 nm was recorded using a Synergy™ Mx microplate reader (BioTek Instruments, Winooski-Vermont-USA) [21]. Briefly, after the first 24 h of incubation, the cell viability assay was performed with CCK-8 at T0 in two wells from each experimental group. Next, 10 μL of CCK-8 solution was added to each well of the plates together with Gibco™ DMEM complete FBS (10% FBS, 1% penicillin/streptomycin, 1% glutamine) (ThermoFisher Scientific, Madrid, Spain). Subsequently, they were kept in the incubator at 37 °C for 45 min. Finally, absorbance was measured using a Synergy™ Mx microplate reader (BioTek Instruments, Winooski-Vermont-USA) at 450 nm. After ending the CCK-8 assay at T0, the study groups were maintained in culture on their respective plates for 3 h and 6 h at 37 °C in the HeraCell™ 150i CO_2_ incubator (Thermo Scientific™, Waltham, MA, USA), with six replicates per experimental group [21]. The absorbance of formazan was measured at 450 nm using the plate reader mentioned above. The percentage (%) of cell viability was calculated as follows [22]:$$\mathrm{Cell}\;\mathrm{viability}\;(\%)=\frac{\mathrm{Absorbance}\;\mathrm{of}\;\mathrm{sample}}{\mathrm{Absorbance}\;\mathrm{of}\;\mathrm{control}\;}\;\times \;100\%$$

### 2.4. Statistical Analysis

Statistical analysis was performed with the SPSS 25.0 program for Windows (IBM, USA). The variable of study was cell viability expressed as the % of cell viability. The normality of the data was analysed using the Shapiro–Wilk test. Student’s *t*-test was used to compare the concentration of MMC released by the BraidStent-SF-MMC at 3 and 6 h. A comparison between groups at 3 and 6 h was carried out using the Kruskal–Wallis test, and, in the case of statistical significance, a corresponding post hoc analysis was carried out using the Bonferroni test. The trend of % of cell viability over time at 3 and 6 h was analysed via the Wilcoxon signed-rank test. The confidence interval set at 95% (95% CI), and significance was determined with *p* < 0.05.

## 3. Results

The results of the artificial urine study for the determination of the concentration of MMC released by the BraidStent-SF-MMC at 3 and 6 h are shown in Figure 2. No significance was found between groups (Student’s test). Regarding the percentage of release with respect to the MMC encapsulated in the stent, 81.7% was released at 3 h, and 100% was released at 6 h. The urine study demonstrated the stability of the SF coating in a physiological environment.

Regarding the results of the cell viability % studies according to the determination of absorbance at T3 and T6, the data do not follow a normal distribution.

As for the comparison of the cell viability % between groups over time, two different trends were observed. On the one hand, a trend of G3 and G4 corresponding to the non-MMC groups with BraidStent-SF and T24 cell culture alone (negative control), and on the other hand, the trend of the cultures in which MMC was present in G1 and G2, with a lower cell viability % with respect to the two control groups (Figure 3).

From the point of view of the tendency over time within each group, we found statistical significance between 3 and 6 h regarding the groups with MMC (G1 and G2), with a decrease in cell viability related to exposure time. In the non-MMC groups, only G3 showed statistical significance over time, increasing its cell viability % (Figure 3). In detail, the cell viability of the T24 cells in the presence of G1 and G2 at 3 h were 62.21 ± 2.04% and 65.40 ± 5.26%, respectively. Additionally, at 6 h, it was 52.29 ± 2.42% and 47.66 ± 3.78%, respectively.

Regarding the inter-group comparison of the percentage of cell viability at T24, G1 showed statistical significance at T3 compared to G3 (*p* = 0.013) and G4 (*p* = 0.001), and G2 only showed statistical significance with G4 (*p* = 0.012). At T6, significance was found for the two groups with MMC, G1, and G2 compared to the two negative controls: the G3 (bare BraidStent-SF) culture and G4 (T24 cell culture alone) (*p* < 0.005) (Figure 3).

## 4. Discussion

The search for new technologies for the instillation of adjuvant chemotherapy in urothelial carcinoma of the upper urinary tract is one of the current challenges in urological oncology due to the shortcomings of current systems [3,7]. Therefore, the development of a ureteral stent that delivers chemotherapy on a scheduled basis may be of great benefit in patients with low-grade UTUC or in those patients who are not suitable candidates for radical nephroureterectomy despite their tumours being high-grade [17].

The main advantages of the development of a biodegradable ureteral stent such as BraidStent-SF-MMC are mainly that it allows the release of anti-cancer drugs such as MMC into the upper urinary tract; this release system allows a longer dwell time of the chemotherapy drug with the urothelium, improving its cytotoxic effect [19]. A second surgical intervention is not required to remove the stent, as it is biodegradable [12,13,14]. Additionally, it is not related to the possible increases in intrarenal pressure that may cause urosepsis, which occurs in cases of intracavitary instillation via ureteral catheters. The latter currently requires prior assessment of renal volume so as not to exceed the intrarenal pressure that could cause pyelovenous or pyelolymphatic reflux, which is a limiting factor in achieving the recommended chemotherapy dose [3,7,9].

The preliminary results confirm that the BraidStent coated with the SF matrix and embedded MMC in its 10 dips (dip-coating technique), called BraidStent-SF-MMC, allows the controlled release of MMC into the urinary environment at 3 and 6 h (Figure 2). In view of these results, future studies will be carried out to determine the release rate at earlier phases, such as 1 h. The aim of the stent is to release the effective dose over a prolonged period of time to increase the dwell time of the MMC with the urothelium.

The objectives of the current study are very straightforward, as we need to assess whether the mitomycin C embedded in the SF matrix layers of the BraidStent-SF-MMC maintain their cytotoxic capacity, as well as to compare its inhibitory effect on cell viability versus the dose administered in patients today. The results of the cell viability study show that the MMC groups (G1, G2) show statistical significance compared to the non-MMC groups (G3, G4) at 6 h (Figure 3).

From the first follow-up, the MMC in T24 cell culture, regardless of whether the release is through direct instillation or whether it is impregnated in a SF matrix (G2), shows cytotoxicity (Figure 3). The results indicate that G3 and G4, the negative control groups, show no significance between the two groups at the two follow-ups. In addition, there was no perceptible cytotoxicity exhibited by the non-MMC BraidStent-SF (G3), ruling out any harmful effects related to the SF matrix, and there was statistical significance compared to G4. A very important finding in this study is that there is no statistical difference between G1 and G2, demonstrating that the concentration of MMC released by the BraidStent-SF-MMC is adequate and comparable to that currently used in patients administered MMC as adjuvant therapy to UTUC (Figure 3). It is important to highlight that we chose a suitable test: CCK8. The main characteristics of this test are that it is an easy and sensitive colorimetric assay for the determination of in vitro cell viability in cytotoxicity assays. It is also true that it is possible to use other types of cell viability assays, such as flow cytometry, which allows for a more specific phenotypic quantitative cell viability analysis. Perhaps the use of more than one cell viability assay in our study would have allowed us to obtain more precise details regarding cell viability.

Since 1993, when silk fibroin was recognised by the US Food and Drug Administration (FDA) as a biomaterial, its use has become extremely widespread due to its interesting characteristics. It has excellent biocompatibility, high strength, mechanical toughness, robust flexibility, high processability, tuneable degradation, ease of processing, and ease acquisition [11,23]. Due to its excellent features, SF has been employed in a wide variety of medical applications, such as in drug-eluting stents [23]. It is the characteristics of its processability that allow the control of the crystalline state, Beta-sheet content, and use in biomaterial-controlled-release applications that regulate the kinetics of the incorporated drugs. The stability of SF coatings can be modified by immersion in methanol, which allows the Beta sheet content to be modified. This is related to the rate of degradation and to the release of the drugs embedded in the SF coating. In our case, we used 10 dip-coating cycles to obtain controlled and early release, with the idea of providing a treatment similar to that currently used in patients but that increases the exposure time of MMC.

On the other hand, the MMC groups showed a significant association with exposure time. The longer the exposure time, the greater the cytotoxic effect. This capacity was more important in the BraidStent-SF-MMC group, but no statistical significance was observed compared to G1, with a cell viability of 47.66% compared to 52.29% for G1 (Figure 3). These findings are very encouraging for the use of this stent in patients, as they confirm that the cytotoxic effects increase when the exposure time of the anti-cancer drugs is increased. Currently, the few clinical groups using the intracavitary instillation of MMC for adjuvant therapy for UTUC and to treat carcinoma in situ only provide this therapy for one hour [7,8]. This limited upper tract dwell time in patients could be the cause of recurrence and progression after UTUC kidney-sparing surgery and is a strength of the stent evaluated in this in vitro study [3,7,9].

Our results are consistent with previous studies using the same T24 cell culture and in which different research groups evaluated biodegradable drug-eluting ureteral stents. For example, Barros et al. evaluated four different anti-cancer drugs (epirubicin, paclitaxel, doxorubicin, and gemcitabine) impregnated via a supercritical fluid CO_2_ technique [19]. Wang et al. also developed an in vitro and animal model study of a new design for gradiently degraded electrospun polyester scaffolds with an epirubicin coating [18]. The three in vitro studies are similar despite the different cytostatics evaluated, mainly because there is currently no consensus as to which is the most suitable—although MMC is the most widely used as the clinical level [8,9]. Regarding the incubation time of stents in cell cultures, different ranges are shown: from 3 to 6 h in our study to 24 h in Wang et al. [18] and from 4 to 72 h in Barros et al. [19]. However, in the assessment of these new stent designs, the viability of cancer cells decreases similarly: 62% to 46% [18], 65% [19], and 65% to 47% in our study. Similar to Barros et al., our study found a greater cytostatic effect in the cytostatic-eluting stent group (G2) than in the groups where the anti-cancer drug was instilled in the wells, possibly due to the longer release effect by the SF matrix [19].

## 5. Conclusions

The coating of a biodegradable ureteral stent with a silk fibroin matrix impregnated in layers of mitomycin C allows the release of the cytostatic in artificial urine. Cell viability studies in a human urothelial carcinoma cell line confirm that mitomycin C embedded in the polymeric matrix does not alter its cytotoxic properties and causes a significant decrease in cell viability at 6 h. These findings could be of great use to decrease the recurrence rate in patients with UTUC.

## Figures and Tables

**Figure 1 polymers-14-04081-f001:**
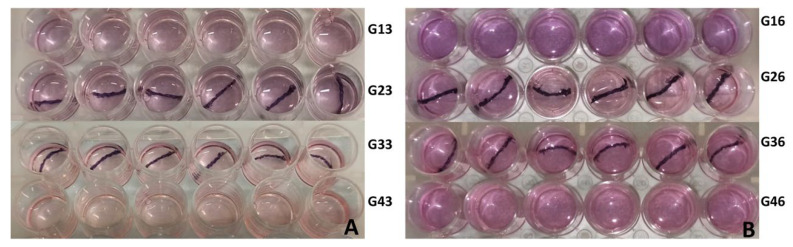
The figure shows the plates that constitute the experimental trial. Each row corresponds to an experimental group. (**A**) refers to T3 (3 h), and (**B**) refers to follow-up at T6 (6 h).

**Figure 2 polymers-14-04081-f002:**
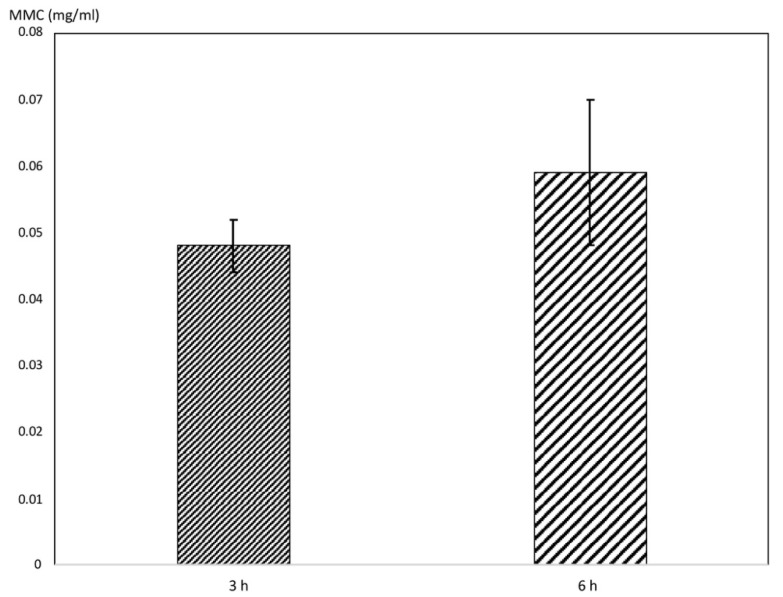
Concentration of MMC (mg/mL) released by BraidStent-SF-MMC at 3 h and 6 h. No significance found between the two groups (Student’s test).

**Figure 3 polymers-14-04081-f003:**
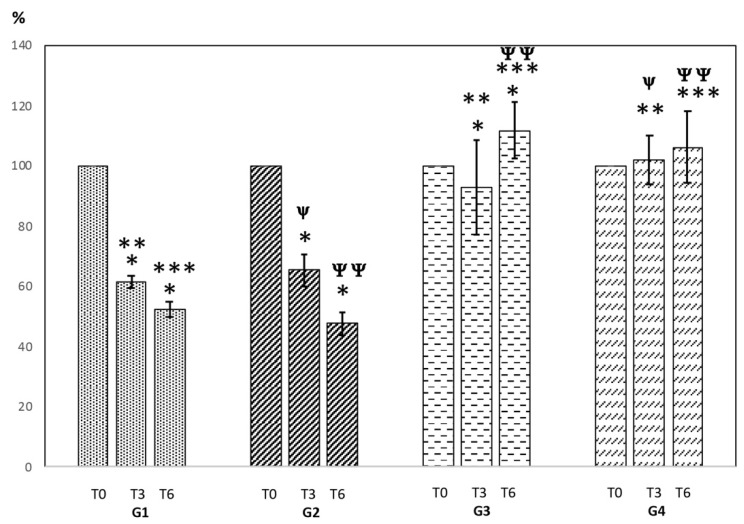
Cell viability % assessment in T24 cell culture (%). T0 (start of study); T3 (3 h); and T6 (6 h). Values indicate mean ± SD. Significance of the values among tested groups within each group and follow-up was determined via a Wilcoxon signed-rank test. The inter-groups at 3 and 6 h were carried out using the Kruskal–Wallis test and Bonferroni post hoc analysis. Statistical significance is depicted as follows: intra-groups (* *p* < 0.05); inter-groups (** *p* < 0.05) (T3: G1 vs. G2, G3, and G4); inter-groups (**^Ψ^** *p* < 0.05) (T3: G2 vs. G3 and G4); inter-groups (*** *p* < 0.05) (T6: G1 vs. G2, G3, and G4), and inter-groups (**^ΨΨ^** *p* < 0.05) (T6: G2 vs. G3 and G4).

**Table 1 polymers-14-04081-t001:** Experimental study groups.

	Groups
G1 (3 h-G13; 6 h-G16)	T24 cell culture + Recommended MMC oce in UTUC (0.66 mg/mL) [8]. No stent. Positive control.
G2 (3 h-G23; 6 h-G26)	T24 cell culture + BraidStent-SF-MMC.
G3 (3 h-G33; 6 h-G36)	T24 cell culture + BraidStent-SF. Negative control.
G4 (3 h-G43; 6 h-G46)	T24 cell culture. No stent. Untreated cells were used as negative control.

## Data Availability

Not applicable.

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
