# Peer review of "Cytotoxicity Assessment of a New Design for a Biodegradable Ureteral Mitomycin Drug-Eluting Stent in Urothelial Carcinoma Cell Culture"

_polymers, 2022, doi:10.3390/polym14194081_

Round 1
Reviewer 1 Report
Soria et al. developed a biodegradable stent for the local delivery of mitomycin C for the treatment of urothelial carcinoma. The cytotoxicity of the materials was evaluated in human cancer cell models using adequate methodologies. The presented study is well-structured, and the discussion is well supported by the obtained results. However, all results relied on data obtained by the CCK-8 proliferation assay, which is quite insufficient to judge the safety and therapeutic efficacy of the developed materials. Additional experiments should be performed to further investigate the safety and efficacy of the developed stent and its effect in the different growth promoting pathways, such as western blot for biomarkers quantification after treatment, or flow cytometry to study drugs’ effect on cell cycle and apoptotic activity. Furthermore, cell studies with a healthy bladder cell line should be also performed to as control.
Additional experiments are strongly recommended and will significantly improve the quality of this manuscript.
Below the authors can find other suggestions and questions:
Keywords: please avoid repeating words from the title, such as drug eluting stent
Line 98: “To determine this concentration,…” should be “To determine MMC release…”
Sections 2.1.3 and 2.1.4 should be combined in a single section
Line 122: the title should be in vitro cytotoxicity instead of ex vivo cytotoxicity
Please include city and country of the manufacture of equipment (e.g. line 137)
Figure 2: It would be more interesting if the authors present the % release regarding the total drug encapsulated in the stent, instead of concentration released.
Regarding the cytotoxicity studies, the IC50 values should be determined and added to the manuscript.
Author Response
R1.1. Thank you very much for your very helpful comments. We fully understand your comments on our manuscript. The overall aim of our experimental study is to advance the development of a biodegradable chemotherapy-eluting stent for upper urinary tract urothelial tumour. Our objective in this study in cell culture was only to assess whether the Mitomycin released by the ureteral stent has cytotoxic capacity, as this is the first study that combines silk fibroin and Mitomycin, so we needed to know if the MMC maintained its cytotoxic capacity to continue advancing in the design of a chemotherapy-eluting stent in UTUC. In a recent experimental study our group has already demonstrated that the stent releases MMC (Soria F, et al. Assessment of a Coated Mitomycin-Releasing Biodegradable Ureteral Stent as an Adjuvant Therapy in Upper Urothelial Carcinoma: A Comparative In Vitro Study. Polymers 2022).
For this reason, no further techniques have been implemented, as the main aim was to assess only the cytotoxicity of silk fibroin matrix-embedded MMC and in this regard we chose the CCK8 test as it is well suited for our purpose. The effects of MMC on the cell cycle are well known and its clinical application in urothelial cancer patients is now routine. We have not implemented safety studies at this stage of stent development, as MMC is commonly used in patients and silk fibroin has been approved by the FDA as a biomaterial for years.
We understand and appreciate the comment to assess in cell cultures of healthy urothelium, but in this experimental study we do not consider it important, as at this stage of stent development our concern was only the cytotoxic capacity of the MMC coating the stent. In previous studies, and as also included in the current manuscript, HPLC-DAD studies were performed to determine the released MMC, which confirmed to us that the MMC was indeed released without modification of its structure.
We fully understand the reviewer's comments and his/her concerns, but for the purpose of our research we honestly believe that the methodology applied is adequate and as this is a long-term study where we must evaluate many factors to further improve and prove the safety and efficacy of this stent, we will certainly implement many of the steps kindly described by the reviewer.
R1. 2. Keywords: please avoid repeating words from the title, such as drug eluting stent. The keyword "drug eluting stent" has been removed.
R1. 3. Line 98: “To determine this concentration,…” should be “To determine MMC release…”. The change suggested by the reviewer has been made.
R1. 4. Sections 2.1.3 and 2.1.4 should be combined in a single section. We are grateful to the reviewer for his/her insightful comment. The change suggested by the reviewer has been made.
R1. 5. Line 122: the title should be in vitro cytotoxicity instead of ex vivo cytotoxicity. We are grateful to the reviewer for his/her insightful comment. The change suggested by the reviewer has been made.
R1. 6. Please include city and country of the manufacture of equipment (e.g. line 137). The requested information from the reviewer has been included for all equipment used in our study.
R1. 7. Figure 2: It would be more interesting if the authors present the % release regarding the total drug encapsulated in the stent, instead of concentration released. We thank the reviewer for the interesting comment. The explanation for figure 2 showing concentration results is that at the clinical level the dose is currently related to mg/ml instilled in the upper urinary tract. We had therefore considered it more useful to express MMC release in this way. On the other hand, for comparison with the therapeutic dose, it is easier to express both as doses. The HPLC-DAD studies we carry out are performed until the stent is free of MMC and this depends on how we formulate the matrix, in the case of the present study, all the MMC contained in the stent is released within the first 6h, with no MMC found in artificial urine in the HPLC-DAD studies beyond 6h (data not included). Thus, 100% of the MMC contained in the BraidStent-SF-MMC is released at 6h and at 3h this percentage is 81.73%. Following the reviewer's advice, these release percentages have been included in the manuscript.
"Regarding the percentage of release with respect to the MMC encapsulated in the stent, 81.7% was released at 3h and 100% at 6h."
R1. 8. Regarding the cytotoxicity studies, the IC50 values should be determined and added to the manuscript. Unfortunately, in this study we did not perform the calculations for the IC50 determination, as we only exposed the cell culture to the concentration currently released by BraidStent-SF-MMC and at the therapeutic dose.
Reviewer 2 Report
The work contains interesting results on in vitro cytotoxicity assessment of novel ureteral mitomycin drug eluting stent.
Anyway, there are some missing experiments/finding to be included.
Details regarding methods adopted for stent preparation are missing.
SF coating should be analysed in terms of quality and stability. In particular, stability tests should be also adopted to assess the bonding strength between the coating and the substrates. TEM and SEM analyses ahould be included also trying to analyse the cross-sections.
A very similar approach has been already published in a recent work of the same group of authors (Polymers; doi 10.3390/polym14153059) including much more details and long-term analyses.
Author Response
We appreciate the reviewer's comments and understand his/her point of view as this manuscript is the continuation of our line of research and the questions he/she raises are included in our first scientific article describing and characterising the BraidStent-SF-MMC (Soria F, et al. Assessment of a Coated Mitomycin-Releasing Biodegradable Ureteral Stent as an Adjuvant Therapy in Upper Urothelial Carcinoma: A Comparative In Vitro Study. Polymers 2022). The current manuscript aims to advance our knowledge of this stent, as well as to improve its design in order to progress to its evaluation in an animal model. To do so, we needed to know whether the MMC released by the stent had cytotoxic capacity after being encapsulated in the SF matrix. As the first manuscript of the BraidStent-SF-MMC has already been published, we have included references to the characterisation of the stent in the current manuscript for a better understanding of the experimental study.
R2.1. Details regarding methods adopted for stent preparation are missing.
Information referring to our group's previous scientific paper describing the stent preparation method has been included so as not to duplicate information.
R2.2. SF coating should be analysed in terms of quality and stability. In particular, stability tests should be also adopted to assess the bonding strength between the coating and the substrates. TEM and SEM analyses should be included also trying to analyse the cross-sections.
This information has been included in our research group's paper: Soria F, et al. Evaluation of a mitomycin-eluting coated biodegradable ureteral stent as adjuvant therapy in upper urothelial carcinoma: an in vitro comparative study. Polymers 2022. This information is included in the current manuscript (2.1.2. Materials for stent SF and MMC coating).
R2.3. A very similar approach has been already published in a recent work of the same group of authors (Polymers; doi 10.3390/polym14153059) including much more details and long-term analyses.
As the reviewer comments, this current manuscript is a continuation of our line of research on biodegradable chemotherapy-eluting ureteral stents for the treatment of UTUC. In the first scientific paper we evaluated and described the BraidStent-SF-MMC and in the current manuscript our aim is to determine the cytotoxic capacity of the MMC released by the stent. This is a long journey that began in 2016 and is now yielding its first scientific results.
Reviewer 3 Report
The manuscript reported by Soria et al entitled as “CYTOTOXICITY ASSESSMENT OF A NEW DESIGN OF BIODEGRADABLE URETERAL MITOMYCIN DRUG-ELUTING STENT IN UROTHELIAL CARCINOMA CELL CULTURE.” attempts to develop a biodegradable ureteral stent coated with a silk fibroin matrix for the controlled release of mitomycin C as an anti-cancer drug. The approach reported by the authors are interesting, however the extent of characterization performed in the study is not sufficient to completely prove their claims. Hence, this manuscript requires a major revision before the final acceptance. My specific comments about the manuscript are follows.
1. The surface features of the silk fibroin coated, and uncoated stents should be provided. Characterizations like Scanning Electron Microscopy (SEM) should be used to show the uniformity of the coating on the stent surface.
2. There are no results presented related with the coating stability under physiological conditions. Authors should study this in the presence of physiological condition mimicking buffer solutions like PBS.
3. Why authors have selected 3&6h as time points for both MMC release studies and cell viability studies. The release in percentage at 3h is quite high 81.7%. Authors should explain why this phenomenon. In my opinion, authors should include 2 more time points such as 0.5,1 before the 3h. This will give a better understanding of the release profile characteristics.
4. The major claim of the study is the better anticancer capabilities of the MMC incorporated stent. However, the preliminary cell viability results presented in the study is not strong enough to support this claim. Authors should perform biological assays like flow cytometry analysis to obtain more detailed insight on the anticancer potential of their system.
5. Finally, in the methods section the information regarding the biodegradable stent preparation is not sufficient for the successful reproducibility of this work. Hence, authors should expand this section by providing more detailed information regarding the preparation.
………………………………………………………………………………………………………
Author Response
Thank you for your comments concerning our manuscript. Those comments are all valuable and very helpful for revising and improving our paper, as well as the important guiding significance to our researches. We have studied comments carefully and have made correction which we hope meet with approval.
R3.1. The surface features of the silk fibroin coated, and uncoated stents should be provided. Characterizations like Scanning Electron Microscopy (SEM) should be used to show the uniformity of the coating on the stent surface.
R3.1. Authors Reply. Thank you for your very helpful comment. Surface characterisation was performed in our previous paper (Soria F, et al. Assessment of a coated mitomycin-releasing biodegradable ureteral stent as an adjuvant therapy in upper urothelial carcinoma: A Comparative In Vitro Study. Polymers (Basel) 2022;14:3059) using sulforhodamine B (SRB) (a fluorescent aqueous marker) in PBS and urine.
This current manuscript of the BraidStent-SF-MMC evaluation only aims to demonstrate whether the stent-released MMC remains cytotoxic, so an extensive description of the stent characterisation was not done. To improve this weakness, the paragraph containing this information has been revised (2.1.2. Materials for stent SF and MMC coating). Unfortunately, we did not carry out SEM studies as these were done by fluorescence microscopy.
R3.2. There are no results presented related with the coating stability under physiological conditions. Authors should study this in the presence of physiological condition mimicking buffer solutions like PBS.
R3.2. Authors Reply. Thank you for your comment. In the previous study BraidStent-SF-MMC was evaluated first in PBS and then in urine, as it is a urinary stent and in further animal model studies and clinical trials it will degrade in urine [reference 17]. In addition, the study we carried out in this experiment for the determination of MMC release was developed in urine.
A sentence is included in the manuscript in reference to your comments, and the screening of stent degradation in urine [reference 17]. Results: “The urine study demonstrated the stability of the SF coating in a physiological environment”.
Furthermore, in our preliminary studies, we have already evaluated its suitable biocompatibility and lack of side effects in animal models, but these are still preliminary studies.
- Soria, F. et al. Comparative animal model assessment of a drug eluting ureteral stent for adjuvant therapy in upper tract urothelial carcinoma. European Urology Open Science 2022;39:S152.
R3.3. Why authors have selected 3&6h as time points for both MMC release studies and cell viability studies. The release in percentage at 3h is quite high 81.7%. Authors should explain why this phenomenon. In my opinion, authors should include 2 more time points such as 0.5,1 before the 3h. This will give a better understanding of the release profile characteristics.
R3.3. Authors Reply. In our previous studies, where the aim was to characterise the coating and assess whether there was release of MMC from the stent, we found that the main rate of release was between 6 and 12 hours after dipping in urine. Since the stent was designed for the early release of MMC (3-6 hours) and its degradation by hydrolysis to occur in urine.
Therefore, in this current study we included the evaluation at 3 hours, to make a deeper assessment. The time of release can be controlled by manipulation of the crystallisation of the Beta-sheet of the SF by methanol dips, as the final clinical aim is early release (between 3 and 6 hours) after placement in patients. That is the reason behind our efforts to evaluate these times, which are relevant to translate the results to the demands of chemotherapy therapy in patients. Therefore, we are looking for times in which the MMC load is completely released, as this should closely correspond to the “effective dose” of the MMC in patients. In this manuscript, our only objective was to assess the cytotoxicity of the released MMC, to verify that the SF matrix did not alter its cytotoxic properties, so no particular emphasis was given to these considerations. However, in our overall study of stent characterisation and improvement, it is clear that we need to further specify these MMC release characteristics and release rate. Changes have been made to the manuscript to include the reviewer's comments.
Discussion: “In view of these results, future studies will be carried out to determine the release rate at earlier phases, such as 1 hour. Although the aim of the stent is to release the effective dose over a prolonged period of time to increase the dwell time of the MMC with the urothelium”.
R3.4. The major claim of the study is the better anticancer capabilities of the MMC incorporated stent. However, the preliminary cell viability results presented in the study is not strong enough to support this claim. Authors should perform biological assays like flow cytometry analysis to obtain more detailed insight on the anticancer potential of their system.
R3.4. Authors Reply. We are grateful for the reviewer's insightful comments. Our main objective in this study was to assess whether the released MMC retained its cytotoxic capacity, as this was an unresolved question after our first studies. Therefore, the focus is exclusively on this topic, as there is no previous scientific literature that has evaluated SF and MMC release. Unfortunately, we have only assessed cell viability using the CCK8 test, which is specific for this type of assay as demonstrated by other research groups. Certainly, flow cytometry could provide a lot of information on this issue. We are satisfied with the results obtained, as they confirm that the released MMC, despite being encapsulated in the SF coating, shows a significant decrease in cell viability compared to negative controls. Thus, there is no loss of cytotoxic properties despite the SF dip coating techniques used for the manufacture of the BraidStent-SF. Several sentences has been included in the manuscript to show the weakness of not having used flow cytometry or others cell viability assays in our study.
“It is important to highlight that although we have chosen a suitable test such as CCK8. Its main characteristics are that it is an easy and sensitive colorimetric assay for the determination of in vitro cell viability in cytotoxicity assays. It is also true that it is possible to use other types of cell viability assays, such as flow cytometry, which allows a more specific phenotypic quantitative cell viability analysis. Perhaps more than one cell viability assay would allow us to obtain more precise details of cell viability in our study.”
- Cai L, et al. Comparison of Cytotoxicity evaluation of anticancer drugs between real-time cell analysis and CCK-8 method. ACS Omega 2019;4:12036.
- Marinaro F, et al. A fibrin coating method of polypropylene meshes enables the adhesion of menstrual blood-derived mesenchymal stromal cells: A new delivery strategy for stem cell-based therapies. Int J Mol Sci 2021;22:13385.
- Han SB, Shin YJ, Hyon JY, Wee WR. Cytotoxicity of voriconazole on cultured human corneal endothelial cells. Antimicrob Agents Chemother 2011;55:4519.
R3.5. Finally, in the methods section the information regarding the biodegradable stent preparation is not sufficient for the successful reproducibility of this work. Hence, authors should expand this section by providing more detailed information regarding the preparation.
R3.5. Authors Reply. Thank you very much for your comment. We had referenced it to our previous studies [12-14], but following your comment a brief description has been included to facilitate reproducibility of the biodegradable stent (BraidStent®).
“Two 0.17 mm threads of Glycomer™ 631 and two 0.14 mm threads thick of PGA, all 10 mm long, were used to manufacture the stent. The threads were then braided together to constitute the central core of the stent”.
Reviewer 4 Report
The submitted manuscript has been significantly revised by the authors and the changes are acceptable for further proceedings. I recommend the acceptance of the manuscript in the present form.
Author Response
We thank the reviewer for his/her work in reviewing our manuscript and for considering our scientific paper positively for publication.
Reviewer 5 Report
This manuscript describes a CYTOTOXICITY ASSESSMENT OF A NEW DESIGN OF BIODEGRADABLE URETERAL MITOMYCIN DRUG-ELUTING STENT IN UROTHELIAL CARCINOMA CELL CULTURE. I regret to inform you that I must reject this paper for further consideration for publication for the following reasons:
1. I didn't find any novelty in this manuscript
2. Introduction doesn't provide sufficient background about the title.
3. Research design is inappropriate.
4. Conclusions doesn't supported by the results
Author Response
R5. This manuscript describes a cytotoxicity assessment of a new design of biodegradable ureteral mitomycin drug-eluting stent in urothelial carcinoma cell culture. I regret to inform you that I must reject this paper for further consideration for publication for the following reasons:
Authors: We thank the reviewer for his/her work in reviewing our manuscript. We have studied comments carefully and have made correction which we hope meet with approval.
R5.1. I didn't find any novelty in this manuscript.
R5.1. Authors Reply. Regarding his/her first comment, we indeed find a great novelty in our study. It is the first time that SF has been used in the coating of a urinary stent as a drug-carrier and delivery system of a chemotherapeutic agent such as Mitomycin. The use of SF allows us not only to carry the MMC but also to control its release rate by adjusting the degree of crystallisation of the SF beta-sheet. This is the reason that makes the current manuscript so important, as this is the first time this combination has been used in a urinary stent, and after our first results ([17]. Soria F, et al. Assessment of a Coated Mitomycin-Releasing Biodegradable Ureteral Stent as an Adjuvant Therapy in Upper Urothelial Carcinoma: A Comparative In Vitro Study. Polymers (Basel). 2022;14:3059), it was necessary to know if the released MMC maintains its cytotoxic properties prior to its assessment in in vivo studies. The results shown in the current study allow us to further advance in the development of a biodegradable MMC-eluting ureteral stent for the adjuvant treatment of upper urinary tract urothelial tumours and thus reduce the high rate of recurrence in patients. This high recurrence rate is mainly due to the ineffectiveness of current intracavitary instillation systems for chemotherapy or immunotherapy. Therefore, we consider our design to be an important novelty and the results of current study confirm that MMC, despite being encapsulated in the fibroin matrix, still retains its cytotoxic capacity.
R5.2. Introduction doesn't provide sufficient background about the title.
R5.2. Authors Reply. Following the reviewer's comments, changes have been made to the introduction to provide more information on the importance of developing a chemotherapy-eluting stent in the adjuvant treatment of UTUC, as well as the benefits of using SF as a polymeric matrix in the coating of drug eluting stents.
R5.3. Research design is inappropriate.
R5.3. Authors Reply.
The experimental study is designed to assess only the effect of MMC released by BraidStent-SF-MMC on cell viability in human urothelial carcinoma cell culture. Three control groups were used to compare with BraidStent-SF-MMC group, one positive and two negative. All assessed in a human urothelial carcinoma cell culture, previously used in other studies to evaluate the effects of chemotherapy in this cancer.
- Barros AA, et al. Drug-eluting biodegradable ureteral stent: New approach for urothelial tumors of upper urinary tract cancer. Int J Pharm 2016;513:227.
- Wang J, Wang G, Shan H, Wang X, Wang C, Zhuang X, Ding J, Chen X. Gradiently degraded electrospun polyester scaffolds with cytostatic for urothelial carcinoma therapy. Biomater Sci 2019 ;7:963-74.
The use of the CCK8 assay is, according to the scientific literature, suitable for assessing cell viability.
- Abidi AH, et al. Phytocannabinoids regulate inflammation in IL-1β-stimulated human gingival fibroblasts. J Periodontal Res. 2022 Sep 7. doi: 10.1111/jre.13050
- Liu J, Du J, Li Y, Wang F, Song D, Lin J, Li B, Li L. Catalpol induces apoptosis in breast cancer in vitro and in vivo: Involvement of mitochondria apoptosis pathway and post-translational modifications. Toxicol Appl Pharmacol. 2022 Sep 3:116215.
- Deng L, Yin Q, Liu S, Luo D. MicroRNA-613Enhances Nasopharyngeal Carcinoma Cell Radiosensitivity via the DNA Methyltransferase 3B/Tissue Inhibitor of Matrix Metalloproteinase-3/Signal Transducer and Activator of Transcription-1/Forkhead Box O-1 Axis. Dis Markers. 2022 Aug 26;2022:5699275.
- Cai L, et al. Comparison of Cytotoxicity evaluation of anticancer drugs between real-time cell analysis and CCK-8 method. ACS Omega 2019;4:12036.
- Marinaro F, et al. A fibrin coating method of polypropylene meshes enables the adhesion of menstrual blood-derived mesenchymal stromal cells: A new delivery strategy for stem cell-based therapies. Int J Mol Sci 2021;22:13385.
It is true that there are other laboratory methodologies for assessing cell viability, and that obviously all have their advantages and disadvantages. Therefore, we consider that our research design follows the standards described by other research groups and allows us to investigate our hypothesis. Following reviewer´s comments we have added the following sentence in the manuscript: “Perhaps more than one cell viability assay would allow us to obtain more precise details of cell viability in our study”.
Although we are certainly aware that additional techniques could have been used in this study, this does not make our experimental design inappropriate in our opinion.
R5.4. Conclusions doesn't supported by the results.
R5.4. Authors Reply. The conclusions we show in our manuscript are:
- “The coating of a biodegradable ureteral stent with a silk fibroin matrix impregnated in layers of mitomycin C allows the release of the cytostatic in artificial urine”. This first sentence corresponds to the findings described in Table 2. Corresponding to: Concentration of MMC (mg/ml) released by BraidStent-SF-MMC at 3 h and 6 h. No significance found between the two groups (Student´s t ).
- “Cell viability studies in a human urothelial carcinoma cell line confirm that mitomycin C embedded in the polymeric matrix does not alter its cytotoxic properties and causes a significant decrease in cell viability at 6 hours”. This second conclusion is supported by the results of the cell culture study shown in Figure 3. Cell viability % assessment in T24 cell culture. We found that study Group 2 showed statistical significance versus the two negative control groups (G3&G4). Furthermore, it showed no statistical difference compared to Group 1, the positive control group. These results support this conclusion.
- “These findings could be of great use to decrease the recurrence rate in patients with UTUC by increasing the dwell time of anti-cancer drugs”. Following the reviewer's suggestions, this sentence has been modified in the manuscript. “These findings could be of great use to decrease the recurrence rate in patients with UTUC”.
Reviewer 6 Report
The authors present a nice work where a biodegradable ureteral stent coated with the silk fibroin matrix and embedded mitomycin C could release the cytostatic in artificial urine. Cell viability studies confirm that the stent does not alter its cytotoxic properties and causes a significant decrease in cell viability at 6 hours. This stent could be useful to treat Urothelial Carcinoma recurrence. I would recommend accepting the paper.
Author Response

(The authors gave the same response as above.)

Round 2
Reviewer 1 Report
The authors addressed all may comments.
Author Response
We thank the reviewer for his/her work in reviewing our manuscript and for considering our scientific paper positively.
Reviewer 3 Report
The Manucript can be accepted in the present format
Reviewer 5 Report
Accpeted